# Designing and Implementing a Sustainable Cooperative Learning in Physical Education: A Pre-Service Teachers' Socialization Issue

**Pascal Legrain \*, Tania Becerra-Labrador, Lucile Lafont and Guillaume Escalié**

Laboratoire Cultures, Education, Sociétés (EA, 7437), University of Bordeaux, 33076 Bordeaux, France;
tania.becerra-labrador@u-bordeaux.fr (T.B.-L.); lucile.lafont@u-bordeaux.fr (L.L.);
guillaume.escalie@u-bordeaux.fr (G.E.)
\* Correspondence: pascal.legrain@u-bordeaux.fr

**Abstract:** The socialization of pre-service teachers (PSTs) depends on various actors. Researchers help them to build knowledge about variables that impact teaching models, including cooperative learning (CL). School teachers help them to efficiently implement teaching–learning environments, including CL configurations in real classrooms. However, these two tutors are insufficiently related to the aim of assisting novice physical education (PE) teachers to play a pivotal role in the transition to sustainable CL practices. Insufficient opportunities are provided for helping PE-PSTs to consider instructional precautions coming back on the theoretical foundations and practical barriers to CL implementation. Therefore, our purpose is to examine the conditions in which synergy between research and professional training may be strengthened to prepare PE-PSTs to durably establish CL in school curricula. The threefold aim of this paper is to examine whether PE-PSTs may be: (a) involved in research for opening new avenues in conducting their project under the researcher's supervision in four main perspectives of CL, (b) trained in CL designs while experiencing instructional approaches and developing competencies to cope with constraints on information sharing, and (c) professionally socialized through the relevant connection between research and applied practice for progressively accessing a realistic and sustainable vision of CL.

**Keywords:** pre-service teacher; professional socialization; theoretical and empirical advances of cooperative learning; empirical research-based approach

## 1. Introduction

During the 20th century, human beings learned that individuals are economically, psychologically, and socially interdependent. This awareness, related to a new reflection on the Anthropocene era, still has various impacts on individual and collective behaviors related to ecological environments. In education, this awareness has recently influenced teaching programs in life sciences and humanities and social sciences, and has been mainstreamed in teacher education. At the same time, these institutional changes in the curriculum provided little room for changes in instructional practices dedicated to in-service and pre-service teacher (PST) education programs. Such a situation questions how social challenges find their way into the school system through the lens of professional development. According to the occupational socialization theory, "The process whereby the individual becomes a participating member of the society of teachers" [1] comprises three phases along a time-oriented continuum (acculturation, professional socialization, and organizational socialization). In PST professional socialization, building knowledge and competencies to help pupils access the complex environment in which they live is just a first step. Acculturation to pedagogical models that lead children to feel positively interdependent to act cooperatively in a social context, and involving cognitive, affective, and motivational resources, is not just a spiritual supplement. It is one of the main competencies that may be developed during the action, on the basis of experience and reflection [2,3].

Personal commitment and positive interdependence are at the heart of cooperative learning (CL), in which the success of individuals is dependent on the success of the group. How this idea filters lastingly into educational practices partly depends on teachers' professional preparation and continuing development. Nevertheless, current data show that the sustainability challenge is still ahead of us. Pianta et al. (2007) [4] showed that pupils only spent less than 5% of the time in small-group instruction at a US elementary school. Using the Cooperative Learning Implementation Questionnaire to examine teacher resistance to implementing CL as an educational innovation, ref. [5] showed that only 15% of the teachers integrated CL designs in their teaching practices. More recently, ref. [6] Buchs, Filippou, Pulfrey as an instructional strategy occasionally, and only 33% used it routinely. In physical education (PE), despite widespread agreement on the importance of developing social skills, there has been little research on cooperative learning and its impact on teacher socialization [7]. The implementation of sustainable CL environments needs coherence and continuity in teacher training in which PSTs hold a key position in the teaching chain. Will the 21st century be a new period for considering CL as a fruitful training environment in which PE-PSTs will have the opportunity learn to teach CL, in turn allowing students to fulfill personal accountability and help each other to learn in a positive, interdependent way?

The aim of this work is to present three main foundations on which a growth perspective of theoretical and practical knowledge of CL can be conceptualized and implemented through PE-PST professional development.

Because the professional socialization process [8] contains a research component, the first vector of a sustainable CL research program refers to the opportunities PE-PSTs would find to target their research project on this topic during their training.

Because the professional socialization process is also centered on teaching models, the second vector concerns the way CL designs would be integrated in PE-PSTs' vocational training to master instructional competencies turned towards a sustainable implementation of CL configurations in classrooms.

Because PE-PSTs are expected to form links between university courses and field-placement experiences, the third vector concerns the bridges that teachers/researchers and instructors can build between theoretical and didactic advances for reinforcing university/school partnerships in teacher education through CL.

## 2. Promoting Sustainable Research Programs on CL for Pre-Service Teachers' Socialization

Cooperative learning is widely recognized as a pedagogical practice that promotes students' socialization and learning at various levels of schooling and across different subject domains [9]. In PE, teachers may seize various targeted opportunities to involve students working together to achieve a common goal that would remain inaccessible without a clear distribution of roles in teams. With respect to this specific context, the traditional pedagogical treatment of sports and physical activities is sparsely influenced by a cultural approach focused on the social–psychological development of children. Although educational policies highlight the crucial interest in social-competence development at school, few pedagogical innovations are centered on pedagogical content knowledge related to the educational benefits of CL. There is still a need to consider whether the organization of structured group work is not only rooted in the official rules of sports, but far more in a large body of pedagogical situations in which children learn to help each other to perform. However, in an educational field based on the social reference of sport, the comprehension of the determinant of cooperation needs to be approached with great caution. From this point of view, PE-PSTs are invited to develop a critical position in considering the factors that block access to expected behaviors that are needed when students are involved in group tasks. Based on the social constructivist theory [10,11], theoretical approaches of CL were mainly conducted under four main perspectives (developmental, cognitive, motivational, and social) that led to unrelated, even contradictory conclusions [12]. These

theoretical orientations serve as a baseline of research acculturation for teachers along a training process in basic and continuing education.

### 2.1. Fostering Development through CL in Childhood

The developmental perspective questions the way pupils are trained to cooperate, and whether the evaluation considers students' behaviors. On one hand, although CL is said to help learners to engage in some manner of cognitive restructuring of new materials in order to learn them, few research protocols have examined whether the maturity of cognitive processes would influence the children's competencies to interact efficiently in groups with respect to their age and education level in the PE domain. Several peer learning configurations invite children to observe teammates provide information and incentives in order to help them to learn and to reach the common goal [13]. According to sub-processes (attention, retention, production, and motivation) embedded in the observational learning, developmental considerations [14] need to be considered to understand the difficulties children face in CL environments. In this theoretical approach, the examination of the environment created by the teacher in helping children to symbolically code modeled actions and translate them into successful behaviors contributing to the team outcomes is relatively undocumented. Working in groups requires children to mobilize cognitive resources for obtaining behavioral changes and performing, but also to relate positively to teammates' activities. This twofold role suggests the teacher is alert to a potential cognitive concurrence of these two orientations that would lead to overloading. This also suggests teachers should deeply scaffold the children's attention and retention processes to scrupulously share the time for individual action mastery, collective action arrangements, and consultation phases around group organization concerns. Although CL conditions in PE emphasize cognitive and physical abilities, as well as the motivational characteristics of learners, this theoretical approach could easily incorporate age-related considerations to investigate different steps in children's acculturation to CL. Implementing cooperative designs at primary and secondary schools raises specific problems. Primary education suggests that teachers deeply consider the students' egocentric tendency and individualistic dispositions for selecting content knowledge and speech acts that emphasize the common goal "we" reach. Secondary education implies middle and high school teachers develop their repertoire of competencies to face antisocial behaviors for preserving and structuring peer-oriented energies of adolescents [15].

The first results of doctoral work in process [16] showed that PE teachers in primary schools are little concerned with the pedagogical precautions that would be taken in order to help children to face their concurrently cognitive orientations. Through the qualitative examination of teacher interviews, the results of Becerra-Labrador and Legrain were consistent with Abramczyk and Jurkowki's (2020) [17] conclusion that teachers are interested in learning more about cooperative learning to use it lastingly in classrooms. Specifically, teachers expressed a lack of information for relevantly scaffolding students' behaviors and help them to cooperate with the school curriculum with respect to age-related changes that impact their needs. Further research conducted from a developmental perspective through a longitudinal model is needed, stressing that managing sustainable CL environments implies continuity along with the curriculum in relation to children's and teenagers' capabilities. This topic, which is related to a sustainable view of CL integrated into the school curriculum, would be more present and visible in the professional socialization process, inspiring the research projects PE-PSTs need to finalize at the end of their vocational training.

### 2.2. Adjusting Cognitive Demands of CL and Promoting Children's Goal Orientations

The cognitive perspective focuses on the learning processes involved in situations in which students ask to make the content understandable to them before making sense to others, helping team members to acquire knowledge and skills. This theoretical perspective was mainly developed by studying CL designs comprising peer learning strategies

according to the old saying "teaching is learning twice". Accordingly, when students were trained to assist teammates, they demonstrated more adjusted self-efficacy appraisals relative to their own motor abilities compared to untrained peer tutors [13]. With regard to sustainable learning processes for making progress, structured peer-learning procedures could favor the accuracy of self-efficacy appraisals in relation to metacognitive awareness, rather than more risky expectations of success [17]. Beyond this evidence, recent research has shed light on new considerations relative to the cognitive characteristics of CL [18]. Specifically, students participate in group work come in with their own academic and social goals [19]. Whether these achievements and social orientations are consistent with the elaboration of a common goal is still understudied.

With regard to task involvement, the emphasis is placed on exerting effort, experiencing improvement, and mastery. With regard to ego involvement, competence and subjective success are tied to the demonstration of superiority. According to the dichotomous model of goal orientations [20–22], some students enter into group work with a view toward making progress and harbor mastery goals while others are concerned about social comparison, and harbor performance goals. The students' academic orientation may have an impact on the investment, particularly on their willingness to help teammates to reach their goals. More recently, the quadratic perspective [23] of achievement goals emphasized four main behavioral investments among students. Among achievement goal orientations, some students contribute to the group work with mastery-approach goals (i.e., doing well relative to task demands or one's own performance trajectory), whereas others express performance-approach goals (i.e., doing well relative to others). Beyond these two achievement orientations, other students' participation is driven by mastery-avoidance goals (i.e., not doing poorly relative to the task demands or one's own performance trajectory) or performance-avoidance goals (i.e., not doing poorly relative to others). Such advances in the achievement goal theory need to be further considered to examine whether all students involved in cooperative designs tend to focus on their efforts, while positive effects derive from working successfully with others [24]. This quadratic perspective should be relevantly integrated into further studies on CL that consider whether mastery-approach goals would be the achievement orientation to favor in helping students to perform cooperatively. Mainly in evaluative situations where each students' motor-skill level is visible within their group and between groups, the literature indicated that students with mastery-avoidance, performance-approach, and performance-avoidance goals would be particularly affected by worry, somatic tension, and bodily symptoms [25].

In achieving goals, students implicitly express social reasons for trying to succeed academically. The literature emphasized that two social goals intrinsically related to achievement goals influence students' attitudes, particularly in how they seek help within group work [19]. Some students' attitudes are driven by social-relationship goals, which refer to an individual desire to form and maintain positive peer relationships in school. Others are more organized by social-responsibility goals in relation to a desire to adhere to social rules and role expectations [26]. According to Cecchini et al. (2011) [27], this issue, which emphasizes the various behavioral responses to help-seeking and offers of assistance, has not been deeply considered in the PE domain. According to academic and social skills embedded in CL [28], PE-PST research papers would be centered on the academic and social goals with which students involve personal accountability and interpersonal skills within face-to-face interactions and group processing, supporting lasting group-working skills in CL environments.

### 2.3. Providing a Motivational Climate in CL

The motivational perspective is centered on the benefits students obtain from the opportunities to make decisions and take responsibilities in small groups. Contrary to teaching configurations that may facilitate the emergence of controlled goal motives based on a competitive reward structure, CL environments are viewed as relevant pedagogical environments that favor autonomous goal motives and emphasize the positive interdepen-

dence and psychological well-being of group members [29]. Nevertheless, beyond the naïve representation of CL as spontaneously motivating, recent studies [30] examined whether a motivational climate (mastery vs. performance) could be attributed to the instructional models that comprise CL. This has been mainly conducted through the examination of task structures, and more recently, through the motivational climate that teachers establish using verbal persuasion to help students make lasting commitments to cooperation [31]. In light of self-determination theory (SDT) [32], cooperative tasks associated with a mastery orientation were considered as relevant learning conditions for nurturing students' psychological needs (competence, autonomy, relatedness) in physical tasks [33]. Recently, Morgan (2019) [34] suggested applying the mastery TARGET structures (task, authority, recognition, grouping, evaluation, and time) [24] to the CL model in PE [35]. The author recommended examining whether students would be: (a) encouraged to set their own self- or group-referenced goals for improvement, (b) involved in decision making and shared leadership, (c) polled in heterogeneous groups, (d) provided with sufficient individual private feedback, (e) taught to use peer-assessment strategies, and (f) placed in flexible time conditions to accommodate the learning needs according to an inclusive perspective. All these new avenues are promising routes for PE-PSTs who would opportunely study the motivational climate set by PE teachers that supports children's social-competence development. According to this motivational perspective, sustainable CL environments are built to prevent lower-achieving students from being blamed for the failure of the group. More generally, it nurtures satisfaction of needs as a significant source of children's intention and moderate-to-vigorous physical activity [36]. Furthermore, considering that learners may adopt the inferred motivational messages of their teachers, influencing their own motivation and their autonomy-supportive attitude when teaching a peer (i.e., motivation contagion) [37], this purpose is a promising avenue for considering sustainable conditions of CL implementation.

## 2.4. Explicitly Structuring Children's Roles to Reinforce Cohesiveness

The social perspective is centered on the cohesiveness of the group and the relations between team members. This theoretical approach [38,39] is rooted in the principle that students help each other to learn because they care about each group member. Furthermore, CL tasks built in heterogeneous groups favor the performance of students who might otherwise not do well [40]. Beyond these interesting results, little is said about the delimitation of social responsibility comprised in students' roles. However, students are often subject to role ambiguity, which refers to the lack of clear and consistent information regarding the role they are expected to endorse within groups. Although this concept is well known in the sports domain [41,42], it is insufficiently considered in PE settings in which CL is implemented. Because "seating people together and calling them a cooperative group does not make them one" [32] (p. 68), further studies would open new avenues to consider how to structure the classroom in PE (Cohen, 1992). A recent study focused on this well-defined purpose and its practical application in PE [43]. According to the multidimensional construct of cohesion [44], the authors examined whether a traditional circuit training with five gymnastic stations would be more beneficial to perceived cohesion when children were trained to endorse clear roles embedded in the group work configuration in comparison to a free incentive to cooperate. The measures were collected through the "Questionnaire sur l'Ambiance du Groupe" [45], which comprises four dimensions that distinguish between individual attractions and group integration in both task and social cohesion. A learning-type-by-gender interaction effect was observed in the group integration-task variation from pretest to post-test. In the free condition of group organization, females' perceptions significantly decreased, whereas those of females involved in the structured CL condition slightly increased. Furthermore, the partial mediating effect of task cohesion in the relationship between the trained CL condition and performance of new motor tasks confirmed the interest of explicitly designing and structuring CL activities to promote peer interactions that influence the classroom cohesion [46,47]. Moreover, when

structured CL conditions are directly focused on pro-social skills that are explicitly taught, it can be assumed that pupils receive an equal opportunity to participate, whatever their gender and skill level [48], enhancing the positive interdependence among them, especially in facing difficult tasks for which they needed help. According to this gymnastic specificity, CL procedures should be adequate to assist both males and female students in 6th grade to go beyond the reluctance to help each other to learn when they are not relevantly prepared to interact efficiently [49]. More generally, this study confirmed Kirk's (2003) [50] assertions by stressing that pupils pooled in mixed-gender groups for practicing PE exercises need to be accompanied by a tailored pedagogy in order to perceive the class as an authentic cooperative environment. With respect to the male-oriented nature of sports and physical activities, theoretical and empirical approaches devoted to the sustainable development of CL in an inclusive conception of PE cannot be oblivious to this gender issue.

## 3. Promoting Sustainable Professional Training Program on CL for Pre-Service Teacher Socialization

Beyond their research training, PSTs also spend hours in training programs specifically dedicated to professional socialization. One of the main goals of teacher education is to provide PSTs with opportunities to experience various instructional models. In the specific context of PE, they learn that students are confronted with positive, negative, and independent goal interdependence [51] through competitive collaborative situations that deeply influence the dynamics and outcomes of interactions. The traditional pedagogical treatment of sports and physical activities is sparsely influenced by a cultural approach focused on the social–psychological development of children. Although educational policies highlight the crucial interest for the social competence development at school, few pedagogical innovations are centered on pedagogical content knowledge related to the educational benefits of group work. University courses and placement experiences would better help PE-PSTs to fill the gap between what they say they want to do and what they are doing in practice. Integrating CL lastingly into PE teaching and learning requires a twofold objective. First, this implies leading PE-PSTs to become more familiar with CL to develop a reflection on the group work implementation not only in collective sports, but also in a large body of situations in which children learn to help each other to perform. Nevertheless, this acculturation of PE-PSTs requires time and the ability to move beyond an implicit presentation of group-work arrangements. Second, this implies inserting CL practical casework experiences in professional training before asking PE-PSTs to implement CL configurations in PE classrooms.

### 3.1. PE-PST Professional Socialization in Building Functional CL Designs

Textbooks that describe CL designs by providing illustrations and recommendations for implementation in real classrooms might be useful in helping PE-PSTs to structure their interventions. Digital resources, including videos, might also provide efficient support in exemplifying how to determine content knowledge and choose teaching practices according to CL [52]. Moreover, several research designs conducted in real classrooms would be informative, encouraging future teachers to modify traditional group-work arrangements with regard to the principles of CL. Specifically, works centered on the implementation of explicit group-work arrangements seem relevant in stressing the roles students have to endorse under the teacher's supervision in PE. In comparison to a group-work organization in which students were implicitly asked to freely help each other to learn, new specific recommendations were recently provided to help students to work in small groups [53]. Beyond traditional precautions for group-work organization (i.e., security criteria, equity for individual passing, and special timing for reading worksheets and practicing), another timing organization was proposed to train the students in reference to CL. Rather than giving a signal that indicated the clockwise rotation on stations for all the groups, the PE teacher invited half of the students to go to the next area to take the role of tutee, whereas those who stayed in their area assumed a tutor role to welcome and assist teammates. The teacher regularly provided pupils with clues, feedback, and

incentives to use the worksheets, and provided a further assessment along with the PE program. This explicit training condition was introduced using the following instructions: "During the second period, you will practice the task in order to perform gymnastic exercises with ease and to explain and show the difficulties to new practitioners. You will pass by the observational landmark and pay attention to each of the tutees' attempts to use relevant technical comments and incentives in order to help them to perform. Don't forget that after this 7 min session you will become a tutee and will have to accept advice from more expert peers on the new station of the circuit." This procedure was based on the following rules for tutors: (a) describe the worksheet and read the clues stressing for teammates the most important elements for execution; (b) systematically take the first place in the group to demonstrate the exercise; and (c) pass the observational landmark to observe teammates and provide feedback on their performances. The tutees' role comprised: (a) scrupulously observing the tutor's demonstrations; (b) evaluating its results before receiving any additional feedback, and (c) listening to comments in order to correct execution and be prepared to take the tutor's role during the next rotation. When implemented in CL configurations, such pedagogical precautions would have an impact on both pupils' self-efficacy beliefs and motor performance in comparison to an unclear and non-explicated CL configuration. This illustration would nurture PE-PSTs' thinking and approaches to structured peer-learning contexts in which social skills are taught to be transferable from a physical practice to another providing a sustainable perspective to CL configurations in PE.

### 3.2. Enabling PSTs to Experiment with CL Configurations during Training Courses

The current PST training conditions increasingly invite future PE teachers to experiment in peer groups with a variety of pedagogical models along the professional socialization, focusing on considerations for framing and reframing the classroom ecology [54]. Nevertheless, the didactic basis of the micro-system for experiencing instructional approaches is more focused on the competencies PE-PSTs acquire when using direct instruction. This traditional teaching model is grounded in an unambiguous presentation of the curriculum through demonstration, and guided and independent practice on activities directly related to the newly learned material [55]. In this instructional-training context, CL configurations are less studied under a practical teaching viewpoint, leaving little room for reflective skills [56]. In line with this lack, recent studies were conducted in a human anatomy course [57] and PE training program [58] that considered Jigsaw as a two-group configuration that would be relevantly integrated into PSTs' vocational training. In reference to Aronson's historical account [59], Jigsaw is a CL environment in which students structure knowledge and skills in an expert phase before coming back to their initial team to instruct teammates. Legrain et al. (2019) studied whether instructional knowledge explicitly provided in the expert group and Jigsaw group training sessions would influence PE-PSTs' skills, self-efficacy, and knowledge for instruction in contrast to a Jigsaw experience only, and direct instruction. Although no difference was found between the three training conditions on self-efficacy over time, the participants explicitly provided with instructional knowledge in CL scored significantly higher on knowledge for instruction and motor skills than participants of the two other training conditions. The authors advanced that benefits would depend on the content knowledge comprised in acts of teaching the instructor highlighted during the intervention [60] to identify the instructional options for selecting and enacting contents to be taught. The results partly confirmed that integrating a CL training condition within the professional socialization stage was a useful alternative model to direct instruction, helping PSTs master requisite knowledge when discovering new content involved in a future teaching function. Nevertheless, other results collected on PE-PSTs' motivation for CL instruction showed that the participants who experienced the Jigsaw procedure perceived the instructor as more autonomy-supportive compared to those who were prompt to focus on the instructional activity. It was suggested that the explicit training condition led participants to perceive

the instructional precautions modeled by this instructor in a controlling rather than an autonomous way when they returned to their team. Furthermore, stressing the difference of professional efficacy between the instructor and the PE-PST, the explicit instructional design would have been perceived as potentially thwarting. Finally, the focal position occupied by the instructor in this training condition seemed to attenuate the expected vicarious effect that would be provided by the observation of teammates [17].

According to sustainable CL environments in teacher professional development, these results suggest progressively planning the instruction through different periods. Considering that, spontaneously experiencing CL configurations would be the best choice in a first training period to favor the participants' satisfaction of the need for autonomy, and involvement in the teaching-learning roles remains to be determined. As relevant as the meaningful rationale provided by videos and textbooks would be for performing the instructional task, this additional information would be more relevantly delivered in the second phase of PST acculturation. Thus, it would be more appropriate to initially help PE-PSTs to feel personally accountable for their instruction, rather than prematurely providing justifications for instructional choices that could be perceived as work pressures thwarting the perceptions of autonomy support.

### 4. Grounding PE-PST Professional Socialization in the Empirical Research-Based Approach of Sustainable CL

Because the concept of sustainability generally requires coherence and continuity in responsible decision-making and actions, several educational actors need to be adequately involved in this process. Specifically, research and professional competencies would be better connected in the training programs for PSTs. The current situation of PE-PST professional socialization prompts us to make two recommendations related to coherence and continuity that are central issues in nurturing a sustainable vision of CL environments.

#### 4.1. Fostering "In-Class" Research Closer to Real Teaching Practices

The coherence of initiatives for building a sustainable CL environment depends on updated research that continues to explore both the positive consequences and the barriers for implementing CL environments in a PE setting. In several school subjects like PE, applying considerations of CL is still necessary to fill the gap between theory and practice. This situation is related to three main issues. First, the current PST acculturation to CL is most often limited to a theoretical presentation of concepts in lecture halls. Such actions/teaching settings are not sufficient in helping PSTs to make a clear distinction between group work and CL. Second, although teachers consider CL instructional models as useful in favoring students' transversal skill development, they also regularly report that school implementation is time-consuming in terms of planning and preparing the tasks, as well as in assessing academic and cooperative skills. Some teachers do not find, in the system of continuing education, the opportunity to share fruitful knowledge and sample lesson-plan formats for building routines. Third, pedagogical content knowledge related to initial preparation is not really grounded in two main interconnected principles: building academic tasks that favor peer interactions, and helping pupils to share fruitful interactions with their peers [6,61]. Such a preoccupation is not currently based enough on a clear articulation of teaching practices with research programs perceived as far too removed from real-world practice. The theoretical approaches are fragmented and insufficiently related to the potential emergence of various outcomes related to the teaching–learning conditions in classrooms.

Considering whether quantitative and qualitative data can be fruitfully combined in CL environments is still a challenge. In the PE domain, a large scope of research is grounded in experimental methods using questionnaires. Irrespective of high incentives provided by research on teachers' efficacy beliefs and the students' representation of the effectiveness of teaching strategies in CL environments, the measures rarely combine these two forms of data collection. New theoretical approaches centered on the cognitive and motivational basis of CL involve considering teachers' and students' perceptions through methodolo-

gies based on the complementarity of quantitative and qualitative data. This work was conducted in complement to Legrain et al.'s (2019) study to access a more comprehensive view of the process by which PE-PSTs develop their teaching self-efficacy and competencies. For this purpose, self-confrontation interviews were conducted using a procedure generally applied in a research program grounded in cultural anthropology, with a view to formally establish the rules a posteriori applied by each participant [62]. Video training episodes, including the two specific phases of the Jigsaw procedure (expertise acquisition and co-teaching), were used to determine the meaning of the action within the context of the training situation. The processing of qualitative data provided an additional rationale for explaining why Jigsaw participants displayed a greater level of pedagogical content knowledge, but no significant difference in the improvement of teaching self-efficacy in comparison with direct-instruction participants. The self-confrontation interviews revealed that the asymmetric position in which PE-PSTs were placed during the expertise phase of Jigsaw mainly led them to judge their teaching competencies as insufficient compared to those of the instructor [63]. Despite the relevance of such methodologies in grasping the complexity of determinants and consequences of teaching practices built on sustainable CL environments, mixed methods are still rare, as authors generally experience difficulty in negotiating the editorial process to disseminate results of research in professional worlds.

However, the research–practice articulation suggests that educational policies set up a professional training scheme to give teachers and researchers the opportunity to share knowledge around CL issues. The implementation of CL environments is based on good intentions, and is less influenced by a deep examination of relations between theoretical and practical foundations of CL. This often leads teachers to introduce sparse CL environments in the ecology of classrooms, with insufficient benefits to ensure the sustainability of such pedagogical experiences. For instance, teachers regularly report that group-work configurations lead to fruitful interactions within small groups [64,65], but also to conflicts of interest involving leadership and control of group activities. Because group-work arrangements, including CL, generate both expected epistemic conflicts and less-expected social conflicts between students [66], researchers might also be interested in the second behavioral tendency to bridge the knowledge between their research topic and the real world of education. This suggests researchers interact with teachers in selecting measures to assess the impact of CL strategies on changes in classroom practices. This also implies the consideration of feedback that teachers would provide when facing antisocial behaviors related to the barriers that students may experience in CL environments. Ensuring a sustainable CL implementation implies that researchers must provide professionals in education with a clear message: Your comments about your experience are valuable for researchers.

### 4.2. Reinforcing University–School Partnerships in Teacher Education

The continuity of initiatives in building a sustainable CL environment depends on the strategy selected to prepare pre-service teachers to experiment with multi-model-based practices comprising CL configurations during practice sessions and traineeship before implementing pedagogical designs in real classrooms. It also depends on the decompartmentalization of subject areas that favor openness among disciplines and contacts between teachers pertaining to a community of practice. Because PE-PSTs are trained by both academics and primary/secondary school teachers within two distinct environments (universities and fields of practical experience), each of these tutors needs to go beyond the traditional reflective scientist/applied modes to better link their efforts to ensure a sustainable vision of CL. From a professional-socialization viewpoint, opening the doors of schools to researchers and novice teachers is required to help them share knowledge and fill the gap between promising theoretical approaches and cautious implementations of CL environments. With regard to these conditions, a prime objective should be that researchers consider CL as a continually developing project that would inspire PE-PSTs' research studies by integrating updated theoretical perspectives. A second objective aims

to help teachers leave their professional comfort zones and consider the diversity of variables that impact educational contexts. PE-PSTs' privileged position is at the frontier between these two worlds, which need to be better connected. When they are faced with pedagogical constraints of CL implementation, they especially need to clearly identify the potential sources of blockages for making decisions in the classroom, expressing the competencies they develop along with the training program. This brings into question the theoretical and methodological foundations on which introductory research courses are based, and whether lectures and internship experience are related in the PE-PSTs' professional socialization.

These observations refer to the traditional relationship between research and teaching, and influence instructional practices dedicated to professional development. New avenues should be pursued that consider qualitative data in examining whether PE-PSTs are restructuring the classroom step-by-step by implementing CL environments in PE.

## 5. Conclusions

According to Incheon Declaration and Sustainable Development Goal 4 (SDG4)—Education 2030 Framework for Action, one of the main targets is to "substantially increase the supply of qualified teachers, including through international cooperation for teacher training in developing countries" [67]. Although the current conditions of PE-PST professional socialization are being reconsidered to meet the challenges of innovative implementations, multi-level and systemic approaches that consider whether research and PE teaching–learning practices could help with the situation are promising. According to CL, a sustainable perspective of these environments is based on the co-creation of solutions provided cooperatively by faculty, staff, and community members in charge of PSTs' professional socialization, which cannot be managed as a top-down initiative. Promoting sustainable CL practices in education depends primarily on whether collective and well-coordinated supports can be provided to strengthen the PE-PSTs' privileged position in the educational system. With regard to their pivotal position, PE-PSTs would become the new ambassadors of CL by contributing to the extension of initial research works and pedagogical initiatives with realism in the school system. The conditions under which cooperative context may be professionally shared between researchers and teachers is a key element in promoting the active participation of PE-PSTs in this challenge. New avenues might be explored by PE-PSTs through their research projects under the four theoretical perspectives considered in real CL classrooms.

Contributing to a sustainable learning community of CL practices in schools implies that each stakeholder must step out of their comfort zone and remain open to educational partners. Institutional authorities need to be part of the project of change, and play a leadership role. School leaders may usefully instill opportunities to help researchers and teachers from different disciplines to work together. High-school principals especially can promote emulation among teachers, providing opportunities for sharing resources and working together in CL environments dedicated to competence building, and returning to the philosophy of teaching. Furthermore, they may contribute to transformative change by welcoming PSTs to boost innovations.

**Author Contributions:** P.L., T.B.-L., L.L. and G.E. contributed to the conceptualization, methodology and validation stages. The writing-original draft preparation and editing were conducted by P.L. Sustainability Editorial office provided a useful support for visualization, review and editing processes. All authors have read and agreed to the published version of the manuscript.

**Funding:** This research received no external funding.

**Institutional Review Board Statement:** Not applicable.

**Informed Consent Statement:** Not applicable.

**Data Availability Statement:** Exclude this statement.

**Conflicts of Interest:** The Authors declare that there is no conflict of interest.

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
