# Peer review of "Designing and Implementing a Sustainable Cooperative Learning in Physical Education: A Pre-Service Teachers’ Socialization Issue"

_sustainability, doi:10.3390/su13020657_

Round 1
Reviewer 1 Report
Dear authors,
I have read your manuscript with very interest and found your analysis to be extremely accurate and grounded.
I am working in physical education and I do agree with you that CL might result in a 'certain' revolutionary pedagogical strategy to foster all educational domains for the preservive students interested in physical education. Since I distinguish the practice both in the university class domains and in the schools domains, I am missing some analysis on the achievement that CL has on the PSTs while learning between peers and on the professional attitudes and competences a PST can obtain while 'exercising' CL in the primary, or secondary education. I am infering that in the primary and secondary education the dimensions that define CL mig be tunned in terms of the students age, with some tendencies to egocentrism for primary students, to a real identification of CL relationships and dimensions for secondary level students. Therefore, it would be nice if you could add some coment on the application of CL at the diferent levels of education, specialy on promotive interacion, commitment, indivual responsibility, etc. This, I think will add another rich perspective to the manuscript. And second, in some of the paragraphs I am missing some references at the end of phrases to better contextualize your work.
Author Response
|
Reviewer 1 Comments to Author: |
Authors’ revision |
|
I am missing some analysis on the achievement that CL has on the PSTs while learning between peers and on the professional attitudes and competences a PST can obtain while 'exercising' CL in the primary, or secondary education. I am inferring that in the primary and secondary education the dimensions that define CL might be tuned in terms of the students’ age, with some tendencies to egocentrism for primary students, to a real identification of CL relationships and dimensions for secondary level students. Therefore, it would be nice if you could add some comment on the application of CL at the different levels of education, especially on promotive interaction, commitment, individual responsibility, etc. |
In agreement with the reviewer’s recommendations with regard to primary and secondary education (see subchapter 1.1., p. 3, lines 24-30). We provided additional information explaining that “Implementing cooperative designs at primary and secondary schools raises specific problems”. More specifically, we indicated that “Primary education suggests deeply consider the students’ egocentric tendency and individualistic dispositions for selecting content knowledge and speech acts that emphasize the common goal “we” reach”. In comparison, we advanced that “Secondary education implies middle and high school teachers develop their repertoire of competencies to face antisocial behaviors for preserving and structuring peer-oriented energies of adolescents (Slavin, 1996a). The reference was added in the reference list (see p. 13, lines 2-3). |
|
Second, in some of the paragraphs I am missing some references at the end of phrases to better contextualize your work. |
As recommended, we indicated references in the following parts of the manuscript: Subchapter 2.2., p. 4, line 10: The following reference (Bandura, 1997) was added with regard to the concept of the accuracy of self-efficacy appraisals. Subchapter 2.2., p. 4, line 13. We referred to the concepts of academic and social goals to Ryan, Hicks, and Midgley (1997). |

Reviewer 2 Report
The analyzed topic is contemporary, significant, and relevant. The article analyses a significant phenomenon in the educational practice.
The keywords are not enough related to the topic.
I would strongly recommend correcting the title by including the mentioned keywords pre-service teacher socialization.
It is also necessary to provide the concept and the content for the key terms professional socialization and/or teacher socialization what are used almost in every paragraph. The professional development process is also mentioned, although the correlation between it and teacher socialization should be justified more.
Sustainable Cooperative Learning is also analyzed in the article, however, it's not clear what makes this cooperative learning sustainable, because the term sustainable is rarely mentioned in the article, which is in general limited to the analysis of the cooperative learning process.
I would recommend justifying this aspect in the conclusions of the article more. I have lacked the justification of the methodology in the article.
It is not clear what research method (data analytical method and/or technique) has been applied to achieve the goal, which is certainly broadly formulated. It is necessary to describe in more detail what was covered by the empirical research-based approach, and how it was implemented. The article has essentially undeniable scientific and practical added value. Distinguishing the limitations of the study and the prospects for further research would enhance the content of the article and its value.
Author Response
|
Dear Editor,
Thank you for giving us the opportunity to revise and resubmit our manuscript re-entitled “Designing and Implementing a Sustainable Cooperative Learning in Physical Education: A Pre-Service Teachers’ Socialization Issue”, according to the Reviewer’s 2 suggestion. As you will see below, we have modified the manuscript taking note of the two reviewers’ comments. We feel that the manuscript has been greatly improved as a result. Please find below our responses. We would like to ask you and the reviewers to accept our most grateful thanks for your constructive remarks and suggestions.
|
||||||||||||||||||||||||||||
